# Proximal Stochastic Methods for Nonsmooth Nonconvex Finite-Sum Optimization

**Sashank J. Reddi**
Carnegie Mellon University
sjakkamr@cs.cmu.edu

**Suvrit Sra**
Massachusetts Institute of Technology
suvrit@mit.edu

**Barnabás Póczos**
Carnegie Mellon University
bapoczos@cs.cmu.edu

**Alexander J. Smola**
Carnegie Mellon University
alex@smola.org

## Abstract

We analyze stochastic algorithms for optimizing nonconvex, nonsmooth finite-sum problems, where the nonsmooth part is convex. Surprisingly, unlike the smooth case, our knowledge of this fundamental problem is very limited. For example, it is not known whether the proximal stochastic gradient method with constant minibatch converges to a stationary point. To tackle this issue, we develop fast stochastic algorithms that provably converge to a stationary point for constant minibatches. Furthermore, using a variant of these algorithms, we obtain provably faster convergence than batch proximal gradient descent. Our results are based on the recent variance reduction techniques for convex optimization but with a novel analysis for handling nonconvex and nonsmooth functions. We also prove *global* linear convergence rate for an interesting subclass of nonsmooth nonconvex functions, which subsumes several recent works.

## 1 Introduction

We study nonconvex, nonsmooth, finite-sum optimization problems of the form

$$\min_{x \in \mathbb{R}^d} \quad F(x) := f(x) + h(x), \text{ where } f(x) := \frac{1}{n} \sum_{i=1}^{n} f_i(x), \tag{1}$$

and each $f_i : \mathbb{R}^d \to \mathbb{R}$ is smooth (possibly nonconvex) for all $i \in \{1, \dots, n\} \triangleq [n]$, while $h : \mathbb{R}^d \to \mathbb{R}$ is nonsmooth but convex and relatively simple.

Such finite-sum optimization problems are fundamental to machine learning when performing regularized empirical risk minimization. While there has been extensive research in solving nonsmooth *convex* finite-sum problems (i.e., each $f_i$ is convex for $i \in [n]$) [4, 16, 31], our understanding of their nonsmooth *nonconvex* counterparts is surprisingly limited. We hope to amend this situation (at least partially), given the widespread importance of nonconvexity throughout machine learning.

A popular approach to handle nonsmoothness in convex problems is via proximal operators [14, 25], but as we will soon see, this approach does not work so easily for the nonconvex problem (1). Nevertheless, recall that proper closed convex function $h$, the *proximal operator* is defined as

$$\text{prox}_{\eta h}(x) := \underset{y \in \mathbb{R}^d}{\operatorname{argmin}} \left( h(y) + \tfrac{1}{2\eta} \|y - x\|^2 \right), \qquad \text{for } \eta > 0. \tag{2}$$

The power of proximal operators lies in how they generalize projections: e.g., if $h$ is the *indicator function* $\mathcal{I}_C(x)$ of a closed convex set $C$, then $\text{prox}_{\mathcal{I}_C}(x) \equiv \text{proj}_C(x) \equiv \operatorname{argmin}_{y \in C} \|y - x\|$.

Throughout this paper, we assume that the proximal operator of $h$ is easy to compute. This is true for many applications in machine learning and statistics including $\ell_1$ regularization, box-constraints, simplex constraints, among others [2, 18].

Similar to other algorithms, we also assume access to a *proximal oracle* (PO) that takes a point $x \in \mathbb{R}^d$ and returns the output of (2). In addition to the number of PO calls, to describe our complexity results we use the incremental first-order oracle (IFO) model.[1] For a function $f = \frac{1}{n} \sum_i f_i$, an IFO takes an index $i \in [n]$ and a point $x \in \mathbb{R}^d$, and returns the pair $(f_i(x), \nabla f_i(x))$.

A standard (batch) method for solving (1) is the proximal-gradient method (PROXGD) [13], first studied for (batch) nonconvex problems in [5]. This method performs the following iteration:

$$x^{t+1} = \text{prox}_{\eta h}(x^t - \eta \nabla f(x^t)), \qquad t = 0, 1, \ldots, \tag{3}$$

where $\eta > 0$ is a step size. The following convergence rate for PROXGD was proved recently.

**Theorem (Informal).** [7]: *The number of IFO and PO calls made by the proximal gradient method (3) to reach $\epsilon$ close to a stationary point is $O(n/\epsilon)$ and $O(1/\epsilon)$, respectively.*

We refer the reader to [7] for details. The key point to note here is that the IFO complexity of (3) is $O(n/\epsilon)$. This is due to the fact that a full gradient $\nabla f$ needs to be computed at each iteration (3), which requires $n$ IFO calls. When $n$ is large, this high cost per iteration is prohibitive. A more practical approach is offered by proximal stochastic gradient (PROXSGD), which performs the iteration

$$x^{t+1} = \text{prox}_{\eta_t h}\left(x^t - \frac{\eta_t}{|I_t|} \sum_{i \in I_t} \nabla f_i(x^t)\right), \quad t = 0, 1, \ldots, \tag{4}$$

where $I_t$ (referred to as minibatch) is a randomly chosen set (with replacement) from $[n]$ and $\eta_t$ is a step size. Non-asymptotic convergence of PROXSGD was also shown recently, as noted below.

**Theorem (Informal).** [7]: *The number of IFO and PO calls made by PROXSGD, i.e., iteration (4), to reach $\epsilon$ close to a stationary point is $O(1/\epsilon^2)$ and $O(1/\epsilon)$ respectively. For achieving this convergence, we impose batch sizes $|I_t|$ that increase and step sizes $\eta_t$ that decrease with $1/\epsilon$.*

Notice that the PO complexity of PROXSGD is similar to PROXGD, but its IFO complexity is independent of $n$; though, this benefit comes at the cost of an extra $1/\epsilon$ factor. Furthermore, the step size must decrease with $1/\epsilon$ (or alternatively decay with the number of iterations of the algorithm). The same two aspects are also seen for *convex* stochastic gradient, in both the smooth and proximal versions. However, in the nonconvex setting there is a key third and more important aspect: *the minibatch size $|I_t|$ increases with $1/\epsilon$.*

To understand this aspect, consider the case where $|I_t|$ is a constant (independent of both $n$ and $\epsilon$), typically the choice used in practice. In this case, the above convergence result no longer holds and it is *not* clear if PROXSGD even converges to a stationary point at all! To clarify, a decreasing step size $\eta_t$ trivially ensures convergence as $t \to \infty$, but the limiting point is not necessarily stationary. On the other hand, increasing $|I_t|$ with $1/\epsilon$ can easily lead to $|I_t| \geq n$ for reasonably small $\epsilon$, which effectively reduces the algorithm to (batch) PROXGD.

This dismal news does not apply to the convex setting, where PROXSGD is known to converge (in expectation) to an optimal solution using constant minibatch sizes $|I_t|$. Furthermore, this problem does not afflict smooth nonconvex problems ($h \equiv 0$), where convergence with constant minibatches is known [6, 21, 22]. Thus, there is a fundamental gap in our understanding of stochastic methods for *nonsmooth nonconvex* problems. Given the ubiquity of nonconvex models in machine learning, bridging this gap is important. We do so by analyzing stochastic proximal methods with guaranteed convergence for constant minibatches, and faster convergence with minibatches independent of $1/\epsilon$.

**Main Contributions**

We state our main contributions below and list the key complexity results in Table 1.

- We analyze nonconvex proximal versions of the recently proposed stochastic algorithms SVRG and SAGA [4, 8, 31], hereafter referred to as PROXSVRG and PROXSAGA, respectively. We show convergence of these algorithms with constant minibatches. To the best of our knowledge, this is the first work to present non-asymptotic convergence rates for stochastic methods that apply to *nonsmooth nonconvex* problems with *constant* (hence more realistic) minibatches.

- We show that by carefully choosing the minibatch size (to be sublinearly dependent on $n$ but still independent of $1/\epsilon$), we can achieve provably faster convergence than both proximal gradient and proximal stochastic gradient. We are not aware of any earlier results on stochastic methods for the general *nonsmooth nonconvex* problem that have faster convergence than proximal gradient.
- We study a nonconvex subclass of (1) based on the proximal extension of Polyak-Łojasiewicz inequality [9]. We show linear convergence of PROXSVRG and PROXSAGA to the optimal solution for this subclass. This includes the recent results proved in [27, 32] as special cases. Ours is the first *stochastic* method with provable global linear convergence for this subclass of problems.

## 1.1 Related Work

The literature on finite-sum problems is vast; so we summarize only a few closely related works. Convex instances of (1) have been long studied [3, 15] and are fairly well-understood. Remarkable recent progress for smooth convex instances of (1) is the creation of variance reduced (VR) stochastic methods [4, 8, 26, 28]. Nonsmooth proximal VR stochastic algorithms are studied in [4, 31] where faster convergence rates for both strongly convex and non-strongly convex cases are proved. Asynchronous VR frameworks are developed in [20]; lower-bounds are studied in [1, 10].

In contrast, nonconvex instances of (1) are much less understood. Stochastic gradient for smooth nonconvex problems is analyzed in [6], and only very recently, convergence results for VR stochastic methods for smooth nonconvex problems were obtained in [21, 22]. In [11], the authors consider a VR nonconvex setting different from ours, namely, where the loss is (essentially strongly) convex but hard thresholding is used. We build upon [21, 22], and focus on handling nonsmooth convex regularizers ($h \not\equiv 0$ in (1)).[2] Incremental proximal gradient methods for this class were also considered in [30] but only asymptotic convergence was shown. The first analysis of a projection version of nonconvex SVRG is due to [29], who considers the special problem of PCA. Perhaps, the closest to our work is [7], where convergence of minibatch nonconvex PROXSGD method is studied. However, typical to the stochastic gradient method, the convergence is slow; moreover, no convergence for constant minibatches is provided.

## 2 Preliminaries

We assume that the function $h(x)$ in (1) is lower semi-continuous (lsc) and convex. Furthermore, we also assume that its domain $\text{dom}(h) = \{x \in \mathbb{R}^d | h(x) < +\infty\}$ is closed. We say $f$ is *L-smooth* if there is a constant $L$ such that

$$\|\nabla f(x) - \nabla f(y)\| \leq L\|x - y\|, \quad \forall\, x, y \in \mathbb{R}^d.$$

Throughout, we assume that the functions $f_i$ in (1) are $L$-smooth, so that $\|\nabla f_i(x) - \nabla f_i(y)\| \leq L\|x - y\|$ for all $i \in [n]$. Such an assumption is typical in the analysis of first-order methods.

One crucial aspect of the analysis for nonsmooth nonconvex problems is the convergence criterion. For convex problems, typically the optimality gap $F(x) - F(x^*)$ is used as a criterion. It is unreasonable to use such a criterion for general nonconvex problems due to their intractability. For smooth nonconvex problems (i.e., $h \equiv 0$), it is typical to measure stationarity, e.g., using $\|\nabla F\|$. This cannot be used for nonsmooth problems, but a fitting alternative is the *gradient mapping*[3] [17]:

$$\mathcal{G}_\eta(x) := \tfrac{1}{\eta}[x - \text{prox}_{\eta h}(x - \eta \nabla f(x))]. \tag{5}$$

When $h \equiv 0$ this mapping reduces to $\mathcal{G}_\eta(x) = \nabla f(x) = \nabla F(x)$, the gradient of function $F$ at $x$. We analyze our algorithms using the gradient mapping (5) as described more precisely below.

**Definition 1.** *A point $x$ output by stochastic iterative algorithm for solving* (1) *is called an $\epsilon$-accurate solution, if* $\mathbb{E}[\|\mathcal{G}_\eta(x)\|^2] \leq \epsilon$ *for some $\eta > 0$.*

Our goal is to obtain *efficient* algorithms for achieving an $\epsilon$-accurate solution, where efficiency is measured using IFO and PO complexity as functions of $1/\epsilon$ and $n$.

| Algorithm | IFO | PO | IFO (PL) | PO (PL) | Constant minibatch? |
|---|---|---|---|---|---|
| PROXSGD | $O\left(1/\epsilon^2\right)$ | $O\left(1/\epsilon\right)$ | $O\left(1/\epsilon^2\right)$ | $O\left(1/\epsilon\right)$ | ? |
| PROXGD | $O\left(n/\epsilon\right)$ | $O\left(1/\epsilon\right)$ | $O\left(n\kappa\log(1/\epsilon)\right)$ | $O\left(\kappa\log(1/\epsilon)\right)$ | – |
| PROXSVRG | $O(n + (n^{2/3}/\epsilon))$ | $O(1/\epsilon)$ | $O((n + \kappa n^{2/3})\log(1/\epsilon))$ | $O(\kappa\log(1/\epsilon))$ | $\checkmark$ |
| PROXSAGA | $O(n + (n^{2/3}/\epsilon))$ | $O(1/\epsilon)$ | $O((n + \kappa n^{2/3})\log(1/\epsilon))$ | $O(\kappa\log(1/\epsilon))$ | $\checkmark$ |

Table 1: Table comparing the *best* IFO and PO complexity of different algorithms discussed in the paper. The complexity is measured in terms of the number of oracle calls required to achieve an $\epsilon$-accurate solution. The IFO (PL) and PO (PL) represents the IFO and PO complexity of PL functions (see Section 4 for a formal definition). The results marked in red are the contributions of this paper. In the table, "constant minibatch" indicates whether stochastic algorithm converges using a constant minibatch size. To the best of our knowledge, it is not known if PROXSGD converges on using constant minibatches for nonconvex nonsmooth optimization. Also, we are not aware of any specific convergence results for PROXSGD in the context of PL functions.

## 3 Algorithms

We focus on two algorithms: (a) proximal SVRG (PROXSVRG) and (b) proximal SAGA (PROXSAGA).

### 3.1 Nonconvex Proximal SVRG

We first consider a variant of PROXSVRG [31]; pseudocode of this variant is stated in Algorithm 1. When $F$ is strongly convex, SVRG attains linear convergence rate as opposed to sublinear convergence of SGD [8]. Note that, while SVRG is typically stated with $b = 1$, we use its minibatch variant with batch size $b$. The specific reasons for using such a variant will become clear during the analysis.

While some other algorithms have been proposed for reducing the variance in the stochastic gradients, SVRG is particularly attractive because of its low memory requirement; it requires just $O(d)$ extra memory in comparison to SGD for storing the average gradient ($g^s$ in Algorithm 1), while algorithms like SAG and SAGA incur $O(nd)$ storage cost. In addition to its strong theoretical results, SVRG is known to outperform SGD empirically while being more robust to selection of step size. For convex problems, PROXSVRG is known to inherit these advantages of SVRG [31].

We now present our analysis of nonconvex PROXSVRG, starting with a result for batch size $b = 1$.

**Theorem 1.** *Let $b = 1$ in Algorithm 1. Let $\eta = 1/(3Ln)$, $m = n$ and $T$ be a multiple of $m$. Then the output $x_a$ of Algorithm 1 satisfies the following bound:*

$$\mathbb{E}[\|\mathcal{G}_\eta(x_a)\|^2] \leq \frac{18Ln^2}{3n-2}\left(\frac{F(x^0) - F(x^*)}{T}\right),$$

*where $x^*$ is an optimal solution of (1).*

Theorem 1 shows that PROXSVRG converges for constant minibatches of size $b = 1$. This result is in strong contrast to PROXSGD whose convergence with constant minibatches is still unknown. However, the result delivered by Theorem 1 is *not* stronger than that of PROXGD. The following corollary to Theorem 1 highlights this point.

**Corollary 1.** *To obtain an $\epsilon$-accurate solution, with $b = 1$ and parameters from Theorem 1, the IFO and PO complexities of Algorithm 1 are $O(n/\epsilon)$ and $O(n/\epsilon)$, respectively.*

Corollary 1 follows upon noting that each inner iteration (Step 7) of Algorithm 1 has an effective IFO complexity of $O(1)$ since $m = n$. This IFO complexity includes the IFO calls for calculating the average gradient at the end of each epoch. Furthermore, each inner iteration also invokes the proximal oracle, whereby the PO complexity is also $O(n/\epsilon)$. While the IFO complexity of constant minibatch PROXSVRG is same as PROXGD, we see that its PO complexity is much worse. This is due to the fact that $n$ IFO calls correspond to one PO call in PROXGD, while one IFO call in PROXSVRG corresponds to one PO call. Consequently, we do not gain any theoretical advantage by using constant minibatch PROXSVRG over PROXGD.

**Algorithm 1:** Nonconvex PROXSVRG $(x^0, T, m, b, \eta)$

---

1: **Input:** $\tilde{x}^0 = x_m^0 = x^0 \in \mathbb{R}^d$, epoch length $m$, step sizes $\eta > 0$, $S = \lceil T/m \rceil$, minibatch size $b$
2: **for** $s = 0$ **to** $S - 1$ **do**
3:     $x_0^{s+1} = x_m^s$
4:     $g^{s+1} = \frac{1}{n} \sum_{i=1}^n \nabla f_i(\tilde{x}^s)$
5:     **for** $t = 0$ **to** $m - 1$ **do**
6:         Uniformly randomly pick $I_t \subset \{1, \ldots, n\}$ (with replacement) such that $|I_t| = b$
7:         $v_t^{s+1} = \frac{1}{b} \sum_{i_t \in I_t} (\nabla f_{i_t}(x_t^{s+1}) - \nabla f_{i_t}(\tilde{x}^s)) + g^{s+1}$
8:         $x_{t+1}^{s+1} = \text{prox}_{\eta h}(x_t^{s+1} - \eta v_t^{s+1})$
9:     **end for**
10:     $\tilde{x}^{s+1} = x_m^{s+1}$
11: **end for**
12: **Output:** Iterate $x_a$ chosen uniformly at random from $\{\{x_t^{s+1}\}_{t=0}^{m-1}\}_{s=0}^{S-1}$.

---

The key question is therefore: *can we modify the algorithm to obtain better theoretical guarantees?* To answer this question, we prove the following main convergence result. For ease of theoretical exposition, we assume $n^{2/3}$ to be an integer. This is only for convenience in stating our theoretical results and all the results in the paper hold for the general case.

**Theorem 2.** *Suppose $b = n^{2/3}$ in Algorithm 1. Let $\eta = 1/(3L)$, $m = \lfloor n^{1/3} \rfloor$ and $T$ be a multiple of $m$. Then for the output $x_a$ of Algorithm 1, we have:*

$$\mathbb{E}[\|\mathcal{G}_\eta(x_a)\|^2] \leq \frac{18L(F(x^0) - F(x^*))}{T},$$

*where $x^*$ is an optimal solution to (1).*

Rewriting Theorem 2 in terms of the IFO and PO complexity, we obtain the following corollary.

**Corollary 2.** *Let $b = n^{2/3}$ and set parameters as in Theorem 2. Then, to obtain an $\epsilon$-accurate solution the IFO and PO complexities of Algorithm 1 are $O(n + n^{2/3}/\epsilon)$ and $O(1/\epsilon)$, respectively.*

The above corollary is due to the following observations. From Theorem 2, it can be seen that the total number of inner iterations (across all epochs) of Algorithm 1 to obtain an $\epsilon$-accurate solution is $O(1/\epsilon)$. Since each inner iteration of Algorithm 2 involves a call to the PO, we obtain a PO complexity of $O(1/\epsilon)$. Further, since $b = n^{2/3}$ IFO calls are made at each inner iteration, we obtain a net IFO complexity of $O(n^{2/3}/\epsilon)$. Adding the IFO calls for the calculation of the average gradient (and noting that $T$ is a multiple of $m$), as well as noting that $S \geq 1$, we obtain a total cost of $O(n + n^{2/3}/\epsilon)$. A noteworthy aspect of Corollary 2 is that its PO complexity matches PROXGD, but its IFO complexity is significantly decreased to $O(n + n^{2/3}/\epsilon)$ as opposed to $O(n/\epsilon)$ in PROXGD.

### 3.2 Nonconvex Proximal SAGA

In the previous section, we investigated PROXSVRG for solving (1). Note that PROXSVRG is not a fully "incremental" algorithm since it requires calculation of the full gradient once per epoch. An alternative to PROXSVRG is the algorithm proposed in [4] (popularly referred to as SAGA). We build upon the work of [4] to develop PROXSAGA, a nonconvex proximal variant of SAGA.

The pseudocode for PROXSAGA is presented in Algorithm 2. The key difference between Algorithm 1 and 2 is that PROXSAGA, unlike PROXSVRG, avoids computation of the full gradient. Instead, it maintains an average gradient vector $g^t$, which changes at each iteration (refer to [20]). However, such a strategy entails additional storage costs. In particular, for implementing Algorithm 2, we must store the gradients $\{\nabla f_i(\alpha_i^t)\}_{i=1}^n$, which in general can cost $O(nd)$ in storage. Nevertheless, in some scenarios common to machine learning (see [4]), one can reduce the storage requirements to $O(n)$. Whenever such an implementation of PROXSAGA is possible, it can perform similar to or even better than PROXSVRG [4]; hence, in addition to theoretical interest, it is of significant practical value.

We remark that PROXSAGA in Algorithm 2 differs slightly from [4]. In particular, it uses minibatches where two sets $I_t, J_t$ are sampled at each iteration as opposed to one in [4]. This is mainly for the ease of theoretical analysis.

---

**Algorithm 2:** Nonconvex PROXSAGA $\left(x^0, T, b, \eta\right)$

---

1: **Input:** $x^0 \in \mathbb{R}^d$, $\alpha_i^0 = x^0$ for $i \in [n]$, step size $\eta > 0$, minibatch size $b$
2: $g^0 = \frac{1}{n} \sum_{i=1}^n \nabla f_i(\alpha_i^0)$
3: **for** $t = 0$ **to** $T - 1$ **do**
4:     Uniformly randomly pick sets $I_t, J_t$ from $[n]$ (with replacement) such that $|I_t| = |J_t| = b$
5:     $v^t = \frac{1}{b} \sum_{i_t \in I_t} (\nabla f_{i_t}(x^t) - \nabla f_{i_t}(\alpha_{i_t}^t)) + g^t$
6:     $x^{t+1} = \text{prox}_{\eta h}(x^t - \eta v^t)$
7:     $\alpha_j^{t+1} = x^t$ for $j \in J_t$ and $\alpha_j^{t+1} = \alpha_j^t$ for $j \notin J_t$
8:     $g^{t+1} = g^t - \frac{1}{n} \sum_{j_t \in J_t} (\nabla f_{j_t}(\alpha_{j_t}^t) - \nabla f_{j_t}(\alpha_{j_t}^{t+1}))$
9: **end for**
10: **Output:** Iterate $x_a$ chosen uniformly random from $\{x^t\}_{t=0}^{T-1}$.

---

We prove that as in the convex case, nonconvex PROXSVRG and PROXSAGA share similar theoretical guarantees. In particular, our first result for PROXSAGA is a counterpart to Theorem 1 for PROXSVRG.

**Theorem 3.** *Suppose $b = 1$ in Algorithm 2. Let $\eta = 1/(5Ln)$. Then for the output $x_a$ of Algorithm 2 after $T$ iterations, the following stationarity bound holds:*

$$\mathbb{E}[\|\mathcal{G}_\eta(x_a)\|^2] \leq \frac{50Ln^2}{5n - 2} \frac{F(x^0) - F(x^*)}{T},$$

*where $x^*$ is an optimal solution of* (1).

Theorem 3 immediately leads to the following corollary.

**Corollary 3.** *The IFO and PO complexity of Algorithm 3 for $b = 1$ and parameters specified in Theorem 3 to obtain an $\epsilon$-accurate solution are $O(n/\epsilon)$ and $O(n/\epsilon)$ respectively.*

Similar to Theorem 2 for PROXSVRG, we obtain the following main result for PROXSAGA.

**Theorem 4.** *Suppose $b = n^{2/3}$ in Algorithm 2. Let $\eta = 1/(5L)$. Then for the output $x_a$ of Algorithm 2 after $T$ iterations, the following holds:*

$$\mathbb{E}[\|\mathcal{G}_\eta(x_a)\|^2] \leq \frac{50L(F(x^0) - F(x^*))}{3T},$$

*where $x^*$ is an optimal solution of Problem* (1).

Rewriting this result in terms of IFO and PO access, we obtain the following important corollary.

**Corollary 4.** *Let $b = n^{2/3}$ and set parameters as in Theorem 4. Then, to obtain an $\epsilon$-accurate solution the IFO and PO complexities of Algorithm 2 are $O(n + n^{2/3}/\epsilon)$ and $O(1/\epsilon)$, respectively.*

The above result is due to Theorem 4 and because each iteration of PROXSAGA requires $O(n^{2/3})$ IFO calls. The number of PO calls is only $O(1/\epsilon)$, since make one PO call for every $n^{2/3}$ IFO calls.

**Discussion**: It is important to note the role of minibatches in Corollaries 2 and 4. Minibatches are typically used for reducing variance and promoting parallelism in stochastic methods. But unlike previous works, we use minibatches as a theoretical tool to improve convergence rates of both nonconvex PROXSVRG and PROXSAGA. In particular, by carefully selecting the minibatch size, we can improve the IFO complexity of the algorithms described in the paper from $O(n/\epsilon)$ (similar to PROXGD) to $O(n^{2/3}/\epsilon)$ (matching the smooth nonconvex case). Furthermore, the PO complexity is also improved in a similar manner by using the minibatch size mentioned in Theorems 2 and 4. [4]

## 4  Extensions

We discuss some extensions of our approach in this section. Our first extension is to provide convergence analysis for a subclass of nonconvex functions that satisfy a specific growth condition popularly known as the Polyak-Łojasiewicz (PL) inequality. In the context of gradient descent,

```
PL-SVRG:(x^0, K, T, m, η)
for k = 1 to K do
   | x^k = ProxSVRG(x^{k-1}, T, m, b, η) ;
end
Output: x^K
```
```
PL-SAGA:(x^0, K, T, m, η)
for k = 1 to K do
   | x^k = ProxSAGA(x^{k-1}, T, b, η) ;
end
Output: x^K
```

Figure 1: PROXSVRG and PROXSAGA variants for PL functions.

this inequality was proposed by Polyak in 1963 [19], who showed *global* linear convergence of gradient descent for functions that satisfy the PL inequality. Recently, in [9] the PL inequality was generalized to nonsmooth functions and used for proving linear convergence of proximal gradient. The generalization presented in [9] considers functions $F(x) = f(x) + h(x)$ that satisfy the following:

$$\mu(F(x) - F(x^*)) \leq \frac{1}{2} D_h(x, \mu), \text{ where } \mu > 0$$

$$\text{and } D_h(x, \mu) := -2\mu \min_y \big[ \langle \nabla f(x), y - x \rangle + \frac{\mu}{2} \|y - x\|^2 + h(y) - h(x) \big]. \tag{6}$$

An $F$ that satisfies (6) is called a $\mu$-PL function.

When $h \equiv 0$, condition (6) reduces to the usual PL inequality. The class of $\mu$-PL functions includes several other classes as special cases. It subsumes strongly convex functions, covers $f_i(x) = g(a_i^\top x)$ with only $g$ being strongly convex, and includes functions that satisfy a optimal strong convexity property [12]. Note that the $\mu$-PL functions also subsume the recently studied special case where $f_i$'s are nonconvex but their sum $f$ is strongly convex. Hence, it encapsulates the problems of [27, 32].

The algorithms in Figure 1 provide variants of PROXSVRG and PROXSAGA adapted to optimize $\mu$-PL functions. We show the following *global* linear convergence result of PL-SVRG and PL-SAGA in Figure 1 for PL functions. For simplicity, we assume $\kappa = (L/\mu) > n^{1/3}$. When $f$ is strongly convex, $\kappa$ is referred to as the condition number, in which case $\kappa > n^{1/3}$ corresponds to the high condition number regime.

**Theorem 5.** *Suppose $F$ is a $\mu$-PL function. Let $b = n^{2/3}$, $\eta = 1/5L$, $m = \lfloor n^{1/3} \rfloor$ and $T = \lceil 30\kappa \rceil$. Then for the output $x^K$ of PL-SVRG and PL-SAGA (in Figure 1), the following holds:*

$$\mathbb{E}[F(x^K) - F(x^*)] \leq \frac{[F(x^0) - F(x^*)]}{2^K},$$

*where $x^*$ is an optimal solution of (1).*

The following corollary on IFO and PO complexity of PL-SVRG and PL-SAGA is immediate.

**Corollary 5.** *When $F$ is a $\mu$-PL function, then the IFO and PO complexities of PL-SVRG and PL-SAGA with the parameters specified in Theorem 5 to obtain an $\epsilon$-accurate solution are $O((n + \kappa n^{2/3}) \log(1/\epsilon))$ and $O(\kappa \log(1/\epsilon))$, respectively.*

Note that proximal gradient also has global linear convergence for PL functions, as recently shown in [9]. However, its IFO complexity is $O(\kappa n \log(1/\epsilon))$, which is much worse than that of PL-SVRG and PL-SAGA (Corollary 5).

**Other extensions:** While we state our results for specific minibatch sizes, a more general convergence analysis is provided for any minibatch size $b \leq n^{2/3}$ (Theorems 6 and 7 in the Appendix). Moreover, our results can be easily generalized to the case where non-uniform sampling is used in Algorithm 1 and Algorithm 2. This is useful when the functions $f_i$ have different Lipschitz constants.

## 5 Experiments

We present our empirical results in this section. For our experiments, we study the problem of non-negative principal component analysis (NN-PCA). More specifically, for a given set of samples $\{z_i\}_{i=1}^n$, we solve the following optimization problem:

$$\min_{\|x\| \leq 1, \ x \geq 0} -\frac{1}{2} x^\top \left( \sum_{i=1}^n z_i z_i^\top \right) x. \tag{7}$$

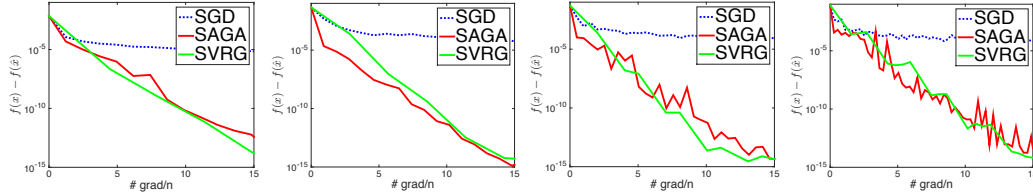

Figure 2: Non-negative principal component analysis. Performance of PROXSGD, PROXSVRG and PROXSAGA on 'rcv1' (left), 'a9a'(left-center), 'mnist' (right-center) and 'aloi' (right) datasets. Here, the y-axis is the function suboptimality i.e., $f(x) - f(\hat{x})$ where $\hat{x}$ represents the best solution obtained by running gradient descent for long time and with multiple restarts.

The problem of NN-PCA is, in general, NP-hard. This variant of the standard PCA problem can be written in the form (1) with $f_i(x) = -(x^\top z_i)^2$ for all $i \in [n]$ and $h(x) = \mathcal{I}_C(x)$ where $C$ is the convex set $\{x \in \mathbb{R}^d | \|x\| \le 1, x \ge 0\}$. In our experiments, we compare PROXSGD with nonconvex PROXSVRG and PROXSAGA. The choice of step size is important to PROXSGD. The step size of PROXSGD is set using the popular $t$-inverse step size choice of $\eta_t = \eta_0(1 + \eta'\lfloor t/n \rfloor)^{-1}$ where $\eta_0, \eta' > 0$. For PROXSVRG and PROXSAGA, motivated by the theoretical analysis, we use a fixed step size. The parameters of the step size in each of these methods are chosen so that the method gives the best performance on the objective value. In our experiments, we include the value $\eta' = 0$, which corresponds to PROXSGD with fixed step size. For PROXSVRG, we use the epoch length $m = n$.

We use standard machine learning datasets in LIBSVM for all our experiments [5]. The samples from each of these datasets are normalized i.e. $\|z_i\| = 1$ for all $i \in [n]$. Each of these methods is initialized by running PROXSGD for $n$ iterations. Such an initialization serves two purposes: (a) it provides a reasonably good initial point, typically beneficial for variance reduction techniques [4, 26]. (b) it provides a heuristic for calculating the initial average gradient $g^0$ [26]. In our experiments, we use $b = 1$ in order to demonstrate the performance of the algorithms with constant minibatches.

We report the objective function value for the datasets. In particular, we report the suboptimality in objective function i.e., $f(x_t^{s+1}) - f(\hat{x})$ (for PROXSVRG) and $f(x^t) - f(\hat{x})$ (for PROXSAGA). Here $\hat{x}$ refers to the solution obtained by running proximal gradient descent for a large number of iterations and multiple random initializations. For all the algorithms, we compare the aforementioned criteria against for the number of *effective* passes through the dataset i.e., IFO complexity divided by $n$. For PROXSVRG, this includes the cost of calculating the full gradient at the end of each epoch.

Figure 2 shows the performance of the algorithms on NN-PCA problem (see Section D of the Appendix for more experiments). It can be seen that the objective value for PROXSVRG and PROXSAGA is much lower compared to PROXSGD, suggesting faster convergence for these algorithms. We observed a significant gain consistently across all the datasets. Moreover, the selection of step size was much simpler for PROXSVRG and PROXSAGA than that for PROXSGD. We did not observe any significant difference in the performance of PROXSVRG and PROXSAGA for this particular task.

## 6 Final Discussion

In this paper, we presented fast stochastic methods for nonsmooth nonconvex optimization. In particular, by employing variance reduction techniques, we show that one can design methods that can provably perform better than PROXSGD and proximal gradient descent. Furthermore, in contrast to PROXSGD, the resulting approaches have provable convergence to a stationary point with constant minibatches; thus, bridging a fundamental gap in our knowledge of nonsmooth nonconvex problems.

We proved that with a careful selection of minibatch size, it is possible to theoretically show superior performance to proximal gradient descent. Our empirical results provide evidence for a similar conclusion even with constant minibatches. Thus, we conclude with an important open problem of developing stochastic methods with provably better performance than proximal gradient descent with *constant minibatch* size.

**Acknowledgment:** SS acknowledges support of NSF grant: IIS-1409802.

---

## Footnotes

[1]Introduced in [1] to study lower bounds of deterministic algorithms for convex finite-sum problems.

[2]More recently, the authors have also developed VR Frank-Wolfe methods for handling constrained problems that do not admit easy projection operators [24].

[3]This mapping has also been used in the analysis of nonconvex proximal methods in [6, 7, 30].

[4]We refer the readers to the full version [23] for a more general convergence analysis of the algorithms.

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
