[Supplementary Material]

# Appendix: Proximal Stochastic Methods for Nonsmooth Nonconvex Finite-Sum Optimization

## A  Convergence analysis for Proximal Nonconvex SVRG

The analysis requires some key lemmas which can be found in Appendix E.

### A.1  General Convergence Analysis

**Theorem 6.** *Suppose $b \leq n^{2/3}$ in Algorithm 1. Let $\eta = b^{3/2}/(3Ln)$, $m = \lfloor n/b \rfloor$ and $T$ be a multiple of $m$. Then for the output $x_a$ of Algorithm 1, the following holds:*

$$\mathbb{E}[\|\mathcal{G}_\eta(x_a)\|^2] \leq \frac{18Ln^2(F(x^0) - F(x^*))}{b^{3/2}(3n - 2b^{3/2})T},$$

*where $x^*$ is an optimal solution of Problem (1).*

*Proof.* We start by defining the full gradient iterate

$$\overline{x}_{t+1}^{s+1} = \mathrm{prox}_{\eta h}(x_t^{s+1} - \eta \nabla f(x_t^{s+1})), \tag{8}$$

which is merely for our analysis, and is never actually computed. Applying Lemma 2 to (8) (with $y = \overline{x}_{t+1}^{s+1}$, $z = x_t^{s+1}$ and $d' = \nabla f(x_t^{s+1})$), and taking expectations we obtain the bound

$$\mathbb{E}[F(\overline{x}_{t+1}^{s+1})] \leq \mathbb{E}\left[F(x_t^{s+1}) + \left[\frac{L}{2} - \frac{1}{2\eta}\right]\|\overline{x}_{t+1}^{s+1} - x_t^{s+1}\|^2 - \frac{1}{2\eta}\|\overline{x}_{t+1}^{s+1} - x_t^{s+1}\|^2\right]. \tag{9}$$

Recall the iterates of Algorithm 1 are computed using the following update:

$$x_{t+1}^{s+1} = \mathrm{prox}_{\eta h}(x_t^{s+1} - \eta v_t^{s+1})), \tag{10}$$

where $v_t^{s+1} = \frac{1}{b}\sum_{i_t \in I_t}(\nabla f_{i_t}(x_t^{s+1}) - \nabla f_{i_t}(\tilde{x}^s)) + g^{s+1}$ (see Algorithm 1). Applying Lemma 2 on update (10) (with $y = x_{t+1}^{s+1}$, $z = \overline{x}_{t+1}^{s+1}$ and $d' = v_t^{s+1}$) and taking expectations we obtain

$$\mathbb{E}[F(x_{t+1}^{s+1})] \leq \mathbb{E}\Big[F(\overline{x}_{t+1}^{s+1}) + \langle x_{t+1}^{s+1} - \overline{x}_{t+1}^{s+1}, \nabla f(x_t^{s+1}) - v_t^{s+1}\rangle$$
$$+ \left[\frac{L}{2} - \frac{1}{2\eta}\right]\|x_{t+1}^{s+1} - x_t^{s+1}\|^2 + \left[\frac{L}{2} + \frac{1}{2\eta}\right]\|\overline{x}_{t+1}^{s+1} - x_t^{s+1}\|^2 - \frac{1}{2\eta}\|x_{t+1}^{s+1} - \overline{x}_{t+1}^{s+1}\|^2\Big]. \tag{11}$$

Adding inequalities (9) and (11), we get

$$\mathbb{E}[F(x_{t+1}^{s+1})] \leq \mathbb{E}\Big[F(x_t^{s+1}) + \left[L - \frac{1}{2\eta}\right]\|\overline{x}_{t+1}^{s+1} - x_t^{s+1}\|^2 + \left[\frac{L}{2} - \frac{1}{2\eta}\right]\|x_{t+1}^{s+1} - x_t^{s+1}\|^2$$
$$- \frac{1}{2\eta}\|x_{t+1}^{s+1} - \overline{x}_{t+1}^{s+1}\|^2 + \underbrace{\langle x_{t+1}^{s+1} - \overline{x}_{t+1}^{s+1}, \nabla f(x_t^{s+1}) - v_t^{s+1}\rangle}_{T_1}\Big] \tag{12}$$

We can bound the term $T_1$ as follows:

$$\mathbb{E}[T_1] \leq \frac{1}{2\eta}\mathbb{E}\|x_{t+1}^{s+1} - \overline{x}_{t+1}^{s+1}\|^2 + \frac{\eta}{2}\mathbb{E}\|\nabla f(x_t^{s+1}) - v_t^{s+1}\|^2$$
$$\leq \frac{1}{2\eta}\mathbb{E}\|x_{t+1}^{s+1} - \overline{x}_{t+1}^{s+1}\|^2 + \frac{\eta L^2}{2b}\mathbb{E}\|x_t^{s+1} - \tilde{x}^s\|^2.$$

The first inequality follows from Cauchy-Schwarz and Young's inequality, while the second inequality is due to Lemma 3. Substituting the upper bound on $T_1$ in (12), we see that

$$\mathbb{E}[F(x_{t+1}^{s+1})] \leq \mathbb{E}\Big[F(x_t^{s+1}) + \left[L - \frac{1}{2\eta}\right]\|\overline{x}_{t+1}^{s+1} - x_t^{s+1}\|^2$$
$$+ \left[\frac{L}{2} - \frac{1}{2\eta}\right]\|x_{t+1}^{s+1} - x_t^{s+1}\|^2 + \frac{\eta L^2}{2b}\|x_t^{s+1} - \tilde{x}^s\|^2\Big]. \tag{13}$$

To further analyze (13), we set up a recursion for which we use the following Lyapunov function:

$$R_t^{s+1} := \mathbb{E}[F(x_t^{s+1}) + c_t \|x_t^{s+1} - \tilde{x}^s\|^2].$$

Introduce the quantities $c_m = 0$, and $c_t = c_{t+1}(1 + \beta) + \frac{\eta L^2}{2b}$. Also, for rest of the analysis set $\beta = b/n$. We can then bound $R_{t+1}^{s+1}$ as follows

$$
\begin{aligned}
R_{t+1}^{s+1} &= \mathbb{E}[F(x_{t+1}^{s+1}) + c_{t+1}\|x_{t+1}^{s+1} - x_t^{s+1} + x_t^{s+1} - \tilde{x}^s\|^2] \\
&= \mathbb{E}[F(x_{t+1}^{s+1}) + c_{t+1}(\|x_{t+1}^{s+1} - x_t^{s+1}\|^2 + \|x_t^{s+1} - \tilde{x}^s\|^2 + 2\langle x_{t+1}^{s+1} - x_t^{s+1}, x_t^{s+1} - \tilde{x}^s\rangle)] \\
&\leq \mathbb{E}[F(x_{t+1}^{s+1}) + c_{t+1}(1 + 1/\beta)\|x_{t+1}^{s+1} - x_t^{s+1}\|^2 + c_{t+1}(1 + \beta)\|x_t^{s+1} - \tilde{x}^s\|^2] \\
&\leq \mathbb{E}\left[F(x_t^{s+1}) + \left[L - \frac{1}{2\eta}\right]\|\overline{x}_{t+1}^{s+1} - x_t^{s+1}\|^2 + \left[c_{t+1}\left(1 + \frac{1}{\beta}\right) + \frac{L}{2} - \frac{1}{2\eta}\right]\|x_{t+1}^{s+1} - x_t^{s+1}\|^2 \right. \\
&\qquad \left. + \left[c_{t+1}(1 + \beta) + \frac{\eta L^2}{2b}\right]\|x_t^{s+1} - \tilde{x}^s\|^2\right] \tag{14} \\
&\leq \mathbb{E}\left[F(x_t^{s+1}) + \left[L - \frac{1}{2\eta}\right]\|\overline{x}_{t+1}^{s+1} - x_t^{s+1}\|^2 + \left[c_{t+1}(1 + \beta) + \frac{\eta L^2}{2b}\right]\|x_t^{s+1} - \tilde{x}^s\|^2\right] \\
&= R_t^{s+1} + \left[L - \frac{1}{2\eta}\right]\mathbb{E}\|\overline{x}_{t+1}^{s+1} - x_t^{s+1}\|^2. \tag{15}
\end{aligned}
$$

The first inequality follows from Cauchy-Schwarz and Young's inequality. The second inequality is due to the bound 13, while the final equality is due to the definition of the Lyapunov function $R_t^{s+1}$. The third inequality holds because the sequence of values $c_t$ satisfies the following bound:

$$c_{t+1}\left(1 + \frac{1}{\beta}\right) + \frac{L}{2} \leq \frac{1}{2\eta}. \tag{16}$$

To verify (16), first observe that $c_m = 0$ and $c_t = c_{t+1}(1 + \beta) + \frac{\eta L^2}{2b}$. Recursing on $t$, we thus obtain

$$
\begin{aligned}
c_t &= \frac{\eta L^2}{2b}\frac{(1 + \beta)^{m-t} - 1}{\beta} = \frac{L}{6b^{1/2}}\left(\left(1 + \frac{b}{n}\right)^{m-t} - 1\right) \\
&\leq \frac{L}{6b^{1/2}}\left(\left(1 + \frac{b}{n}\right)^{\lfloor n/b\rfloor} - 1\right) \leq \frac{L(e - 1)}{6b^{1/2}},
\end{aligned}
$$

where the first equality is due to the definition of $\eta$ and $\beta$, while the first inequality holds because $m = \lfloor n/b\rfloor$. The final inequality follows upon noting that (i) $\lim_{l\to+\infty}(1 + 1/l)^l = e$ and (ii) $(1 + 1/l)^l$ is an increasing function for $l > 0$ (here $e$ is Euler's number). It follows that

$$
\begin{aligned}
c_{t+1}\left(1 + \frac{1}{\beta}\right) + \frac{L}{2} &\leq \frac{L(e - 1)}{6b^{1/2}}\left(1 + \frac{n}{b}\right) + \frac{L}{2} \\
&\leq \frac{Ln(e - 1)}{3b^{3/2}} + \frac{L}{2} \leq \frac{3Ln}{2b^{3/2}} = \frac{1}{2\eta},
\end{aligned}
$$

where the second inequality uses $n/b \geq 1$, while the third inequality uses the hypothesis $n \geq b^{3/2}$. Hence, inequality (16) follows. Now, adding (15) across all the iterations in epoch $s + 1$ and then telescoping sums, we get

$$R_m^{s+1} \leq R_0^{s+1} + \sum_{t=0}^{m-1}\left[L - \frac{1}{2\eta}\right]\mathbb{E}\|\overline{x}_{t+1}^{s+1} - x_t^{s+1}\|^2. \tag{17}$$

Since $c_m = 0$ and from the definition of $\tilde{x}^{s+1}$, it follows that $R_m^{s+1} = \mathbb{E}[F(x_m^{s+1})] = \mathbb{E}[F(\tilde{x}^{s+1})]$. Furthermore, $R_0^{s+1} = \mathbb{E}[F(x_0^{s+1})] = \mathbb{E}[F(\tilde{x}^s)]$. This is due to the fact that $x_0^{s+1} = \tilde{x}^s$. Therefore, using the above reasoning in inequality (17), we have

$$\mathbb{E}[F(\tilde{x}^{s+1})] \leq \mathbb{E}[F(\tilde{x}^s)] + \sum_{t=0}^{m-1}\left[L - \frac{1}{2\eta}\right]\mathbb{E}\|\overline{x}_{t+1}^{s+1} - x_t^{s+1}\|^2. \tag{18}$$

Adding (18) across all the epochs and rearranging the terms, we obtain the bound

$$\sum_{s=0}^{S}\sum_{t=0}^{m-1}\left[\frac{1}{2\eta} - L\right]\mathbb{E}\|\overline{x}_{t+1}^{s+1} - x_t^{s+1}\|^2 \leq F(x^0) - \mathbb{E}[F(\tilde{x}^S)] \leq F(x^0) - F(x^*), \tag{19}$$

where the second inequality follows from the optimality of $x^*$.

Recall that in our notation

$$\mathcal{G}_\eta(x_t^{s+1}) = \tfrac{1}{\eta}[x_t^{s+1} - \text{prox}_{\eta h}(x_t^{s+1} - \eta \nabla f(x_t^{s+1}))] = \tfrac{1}{\eta}[x_t^{s+1} - \overline{x}_{t+1}^{s+1}].$$

Using this relationship in (19) we see that

$$\sum_{s=0}^{S}\sum_{t=0}^{m-1}\left[\tfrac{1}{2\eta} - L\right]\eta^2\mathbb{E}\|\mathcal{G}_\eta(x_t^{s+1})\|^2 \leq F(x^0) - F(x^*). \tag{20}$$

Now using the definition of $x_a$ from Algorithm 1 and simplifying we obtain the desired result. $\quad\square$

**Proof of Theorem 1**

*Proof.* The proof follows from Theorem 6 with $b = 1$. $\quad\square$

**Proof of Theorem 2**

*Proof.* The proof follows from Theorem 6 with $b = n^{2/3}$. $\quad\square$

# B  Convergence analysis for Nonconvex Proximal SAGA

## B.1  General Convergence Analysis

**Theorem 7.** *Suppose $b \leq n^{2/3}$ in Algorithm 2. Let $\eta = b^{3/2}/(5Ln)$. Then for the output $x_a$ of Algorithm 2 after $T$ iterations, the following stationarity bound holds:*

$$\mathbb{E}[\|\mathcal{G}_\eta(x_a)\|^2] \leq \frac{50Ln^2(F(x^0) - F(x^*))}{b^{3/2}(5n - 2b^{3/2})T},$$

*where $x^*$ is an optimal solution of Problem (1).*

*Proof.* We introduce the full-gradient iterate (as before, only for the analysis)

$$\overline{x}^{t+1} = \text{prox}_{\eta h}(x^t - \eta \nabla f(x^t)), \tag{21}$$

and recall that PROXSAGA iterations compute the update

$$x^{t+1} = \text{prox}_{\eta h}(x^t - \eta v^t),$$

where $v^t = \tfrac{1}{b}\sum_{i_t \in I_t}\left(\nabla f_{i_t}(x^t) - \nabla f_{i_t}(\alpha_{i_t}^t)\right) + g^t$. Now, using the same argument as in Theorem 6 until inequality (12), we obtain the following

$$\mathbb{E}[F(x^{t+1})] \leq \mathbb{E}\Big[F(x^t) + \left[L - \tfrac{1}{2\eta}\right]\|\overline{x}^{t+1} - x^t\|^2 + \left[\tfrac{L}{2} - \tfrac{1}{2\eta}\right]\|x^{t+1} - x^t\|^2$$

$$- \tfrac{1}{2\eta}\|x^{t+1} - \overline{x}^{t+1}\|^2 + \underbrace{\langle x^{t+1} - \overline{x}^{t+1}, \nabla f(x^t) - v^t\rangle}_{T_2}\Big]. \tag{22}$$

The term $T_2$ in (22) can be bound as follows:

$$\mathbb{E}[T_2] \leq \frac{1}{2\eta}\mathbb{E}\|x^{t+1} - \overline{x}^{t+1}\|^2 + \frac{\eta}{2}\mathbb{E}\|\nabla f(x^t) - v^t\|^2$$

$$\leq \frac{1}{2\eta}\mathbb{E}\|x^{t+1} - \overline{x}^{t+1}\|^2 + \frac{\eta L^2}{2nb}\sum_{i=1}^{n}\mathbb{E}\|x^t - \alpha_i^t\|^2.$$

The inequality follows from Cauchy-Schwarz and Young's inequality. The second inequality is due to Lemma 4. Substituting the upper bound on $T_2$ in inequality (22), we have

$$\mathbb{E}[F(x^{t+1})] \leq \mathbb{E}\Big[F(x^t) + \left[L - \tfrac{1}{2\eta}\right]\|\overline{x}^{t+1} - x^t\|^2$$

$$+ \left[\tfrac{L}{2} - \tfrac{1}{2\eta}\right]\|x^{t+1} - x^t\|^2 + \frac{\eta L^2}{2nb}\sum_{i=1}^{n}\|x^t - \alpha_i^t\|^2\Big]. \tag{23}$$

For further analysis, we require the following Lyapunov function:

$$R_t := \mathbb{E}\left[F(x^t) + \frac{c_t}{n}\sum_{i=1}^{n}\|x^t - \alpha_i^t\|^2\right].$$

Moreover, for the rest of the analysis we set $\beta = b/4n$. We use $p$ to denote the probability $1 - (1 - 1/n)^b$ of an index $i$ being in $J_t$. Observe that we can bound $p$ from below as

$$p = 1 - \left(1 - \tfrac{1}{n}\right)^b \geq 1 - \tfrac{1}{1+(b/n)} = \tfrac{b/n}{1+b/n} \geq \tfrac{b}{2n}, \tag{24}$$

where the first inequality follows from $(1-y)^r \leq 1/(1+ry)$ (which holds for $y \in [0,1]$ and $r \geq 1$), while the second inequality holds because $b \leq n$.

We now obtain a recursive bound on $R_{t+1}$ as follows

$$R_{t+1} = \mathbb{E}[F(x^{t+1}) + \frac{c_{t+1}}{n}\sum_{i=1}^{n}\|x^{t+1} - \alpha_i^{t+1}\|^2]$$

$$= \mathbb{E}\big[F(x^{t+1}) + \frac{c_{t+1}p}{n}\sum_{i=1}^{n}\|x^{t+1} - x^t\|^2 + \frac{c_{t+1}(1-p)}{n}\sum_{i=1}^{n}\|x^{t+1} - \alpha_i^t\|^2\big]$$

$$= \mathbb{E}\big[F(x^{t+1}) + c_{t+1}p\|x^{t+1} - x^t\|^2$$

$$\quad + \frac{c_{t+1}(1-p)}{n}\sum_{i=1}^{n}(\|x^{t+1} - x^t\|^2 + \|x^t - \alpha_i^t\|^2 + 2\langle x^{t+1} - x^t, x^t - \alpha_i^t\rangle)\big]$$

$$\leq \mathbb{E}\big[F(x^{t+1}) + c_{t+1}\big(1 + \tfrac{1-p}{\beta}\big)\|x^{t+1} - x^t\|^2 + \frac{c_{t+1}(1+\beta)(1-p)}{n}\sum_{i=1}^{n}\|x^t - \alpha_i^t\|^2\big]$$

$$\leq \mathbb{E}\Big[F(x^t) + \big[L - \tfrac{1}{2\eta}\big]\|\overline{x}^{t+1} - x^t\|^2 + \big[c_{t+1}\big(1 + \tfrac{1-p}{\beta}\big) + \tfrac{L}{2} - \tfrac{1}{2\eta}\big]\|x^{t+1} - x^t\|^2$$

$$\quad + \big[\tfrac{c_{t+1}(1+\beta)(1-p)}{n} + \tfrac{\eta L^2}{2nb}\big]\sum_{i=1}^{n}\|x^t - \alpha_i^t\|^2\Big] \tag{25}$$

$$\leq \mathbb{E}\Big[F(x^t) + \big[L - \tfrac{1}{2\eta}\big]\|\overline{x}^{t+1} - x^t\|^2 + \big[\tfrac{c_{t+1}(1+\beta)(1-p)}{n} + \tfrac{\eta L^2}{2nb}\big]\sum_{i=1}^{n}\|x_t - \alpha_i^t\|^2\Big]$$

$$= R_t + \big[L - \tfrac{1}{2\eta}\big]\mathbb{E}\|\overline{x}^{t+1} - x^t\|^2. \tag{26}$$

The equality in the second line follows how $\alpha_i^{t+1}$ is chosen in Algorithm 2. In particular, from noting that each index in $J_t$ is drawn uniformly randomly and independently from $[n]$. The first inequality follows from Cauchy-Schwarz and Young's inequality. The second inequality uses the bound (23). The final equality is due to the definition of the Lyapunov function $R_t$, wherein we also use

$$c_t = \left[c_{t+1}(1+\beta)(1-p) + \frac{\eta L^2}{2b}\right]. \tag{27}$$

The third inequality requires a brief explanation. It follows upon observing that

$$c_{t+1}\left(1 + \frac{1-p}{\beta}\right) + \frac{L}{2} \leq \frac{1}{2\eta}. \tag{28}$$

To see why (28) holds, first observe that $c_T = 0$, and then use (27) to show that

$$c_t \leq c_{t+1}(1-\theta) + \frac{\eta L^2}{2b},$$

where $\theta = (b/2n) - \beta = b/4n$. The above inequality is elementary, since $(1+\beta)(1-p) \leq 1 - p + \beta \leq (1-\theta)$ and because $p \geq (b/2n)$ as noted in (24). Recursing on $t$, we thus obtain

$$c_t \leq \frac{\eta L^2}{2b}\frac{1 - (1-\theta)^{T-t}}{\theta} \leq \frac{2L}{5b^{1/2}}, \tag{29}$$

for all $t \in \{0, \ldots, T-1\}$, which holds due to the definition of $\eta$ and $\theta$. We now use inequality (29) to bound the left hand side of (28) as follows

$$
\begin{aligned}
c_{t+1}\left(1 + \frac{1-p}{\beta}\right) + \frac{L}{2} &\leq \frac{2L}{5b^{1/2}}\left(1 + \frac{2(2n-b)}{b}\right) + \frac{L}{2} \\
&= \frac{2L}{5b^{1/2}}\left[\frac{4n}{b} - 1\right] + \frac{L}{2} \\
&\leq \frac{Ln}{2b^{3/2}}\left[\frac{16}{5} + \frac{b^{3/2}}{n}\right] \leq \frac{5Ln}{2b^{3/2}} = \frac{1}{2\eta}.
\end{aligned}
$$

The first inequality uses the bound (24), while the third inequality uses our hypothesis $b^{3/2}/n \leq 1$. Thus, inequality (28) holds.

Adding the bound (26) across all the iterations and then using telescoping sums, we get

$$
R_T \leq R_0 + \sum_{t=0}^{T-1}\left[L - \tfrac{1}{2\eta}\right]\mathbb{E}\|\overline{x}^{t+1} - x^t\|^2. \tag{30}
$$

Since $c_T = 0$, we observe that $R_T = \mathbb{E}[F(x^T)]$. Furthermore, since $\alpha_i^0 = x^0$ for all $i \in [n]$, we conclude that $R_0 = \mathbb{E}[F(x^0)]$. Therefore, we can rewrite (30) to obtain

$$
\mathbb{E}[F(x^T)] \leq F(x^0) + \sum_{t=0}^{T-1}\left[L - \tfrac{1}{2\eta}\right]\mathbb{E}\|\overline{x}^{t+1} - x^t\|^2.
$$

Rearranging, and using optimality of $x^*$, this leads to the bound

$$
\sum_{t=0}^{T-1}\left[\tfrac{1}{2\eta} - L\right]\mathbb{E}\|\overline{x}^{t+1} - x^t\|^2 \leq F(x^0) - \mathbb{E}[F(x^T)] \leq F(x^0) - F(x^*).
$$

Now recall the relationship

$$
\mathcal{G}_\eta(x^t) = \tfrac{1}{\eta}[x^t - \text{prox}_{\eta h}(x^t - \eta \nabla f(x^t))] = \tfrac{1}{\eta}[x^t - \overline{x}^{t+1}]
$$

and use the definition of $x_a$ (from Algorithm 2) in the above bound to obtain the desired result. $\qquad\square$

### B.2 Proof of Theorem 4

*Proof.* The proof follows from Theorem 7 with $b = n^{2/3}$. $\qquad\square$

## C Convergence Analysis of PL-variants

### C.1 Proof of Theorem 5

*Proof.* The proof follows immediately from Theorem 9 and Theorem 9 with $b = n^{2/3}$. $\qquad\square$

### C.2 PL-SVRG Convergence Analysis

**Theorem 8.** *Suppose $F$ is a $\mu$-PL function. Let $b \leq n^{2/3}$, $\eta = b^{3/2}/(5Ln)$, $m = \lfloor n/b \rfloor$ and $T = \lceil 30Ln/\mu b^{3/2} \rceil$. Then for the output $x^K$ of PL-SVRG, the following holds:*

$$
\mathbb{E}[F(x^K) - F(x^*)] \leq \frac{[F(x^0) - F(x^*)]}{2^K},
$$

*where $x^*$ is an optimal solution of Problem (1).*

*Proof.* We define the following :

$$
\overline{x}_{t+1}^{s+1} = \text{prox}_{\eta h}(x_t^{s+1} - \eta \nabla f(x_t^{s+1})). \tag{31}
$$

We first analyze one iteration of the PROXSVRG for PL functions. PL-SVRG essentially uses this as subroutine multiple times in order to obtain the final solution. The proof is similar to that of Theorem 4 until Equation (11). We have the following inequalities:

$$\mathbb{E}[F(\overline{x}_{t+1}^{s+1})] \leq \mathbb{E}\left[F(x_t^{s+1}) + \left[\frac{L}{2} - \frac{1}{\eta}\right]\|\overline{x}_{t+1}^{s+1} - x_t^{s+1}\|^2\right], \tag{32}$$

$$\mathbb{E}[F(x_{t+1}^{s+1})] \leq \mathbb{E}\Bigg[F(\overline{x}_{t+1}^{s+1}) + \langle x_{t+1}^{s+1} - \overline{x}_{t+1}^{s+1}, \nabla f(x_t^{s+1}) - v_t^{s+1}\rangle$$

$$+ \left[\frac{L}{2} - \frac{1}{2\eta}\right]\|x_{t+1}^{s+1} - x_t^{s+1}\|^2 + \left[\frac{L}{2} + \frac{1}{2\eta}\right]\|\overline{x}_{t+1}^{s+1} - x_t^{s+1}\|^2 - \frac{1}{2\eta}\|x_{t+1}^{s+1} - \overline{x}_{t+1}^{s+1}\|^2\Bigg]. \tag{33}$$

Furthermore, we have the following inequality:

$$\mathbb{E}[F(\overline{x}_{t+1}^{s+1})] \leq \mathbb{E}\left[F(x_t^{s+1}) + \langle\nabla f(x_t^{s+1}), \overline{x}_{t+1}^{s+1} - x_t^{s+1}\rangle + \frac{L}{2}\|\overline{x}_{t+1}^{s+1} - x_t^{s+1}\|^2 + h(\overline{x}_{t+1}^{s+1}) - h(x_t^{s+1})\right]$$

$$\leq \mathbb{E}\left[F(x_t^{s+1}) + \langle\nabla f(x_t^{s+1}), \overline{x}_{t+1}^{s+1} - x_t^{s+1}\rangle + \frac{1}{2\eta}\|\overline{x}_{t+1}^{s+1} - x_t^{s+1}\|^2 + h(\overline{x}_{t+1}^{s+1}) - h(x_t^{s+1})\right]$$

$$= \mathbb{E}\left[F(x_t^{s+1}) - \frac{\eta}{2}D_h(x_t^{s+1}, \frac{1}{\eta})\right] \leq \mathbb{E}\left[F(x_t^{s+1}) - \frac{\eta}{2}D_h(x_t^{s+1}, \mu)\right]$$

$$\leq \mathbb{E}\left[F(x_t^{s+1}) - \mu\eta[F(x_t^{s+1}) - F(x^*)]\right] \tag{34}$$

The first inequality follows from Lipschitz continuity of the gradient of $f$. The second inequality follows from the fact that $\eta < 1/L$. The third inequality follows from the fact that $D_h(x, .)$ is a decreasing function. Here, we are implicitly using the fact that $\mu \leq L$ (which can be shown easily for $\mu$-PL functions that are $L$-smooth). Adding $2/3\times$ Equation (32) and $1/3\times$ Equation (34), we have the following:

$$\mathbb{E}[F(\overline{x}_{t+1}^{s+1})] \leq \mathbb{E}\left[F(x_t^{s+1}) + \left[\frac{L}{3} - \frac{2}{3\eta}\right]\|\overline{x}_{t+1}^{s+1} - x_t^{s+1}\|^2 - \frac{\mu\eta}{3}[F(x_t^{s+1}) - F(x^*)]\right]. \tag{35}$$

Adding the above equation with Equation (33), we have the following:

$$\mathbb{E}[F(x_{t+1}^{s+1})] \leq \mathbb{E}\Bigg[F(x_t^{s+1}) + \left[\frac{5L}{6} - \frac{1}{6\eta}\right]\|\overline{x}_{t+1}^{s+1} - x_t^{s+1}\|^2 + \left[\frac{L}{2} - \frac{1}{2\eta}\right]\|x_{t+1}^{s+1} - x_t^{s+1}\|^2$$

$$- \frac{\mu\eta}{3}[F(x_t^{s+1}) - F(x^*)] - \frac{1}{2\eta}\|x_{t+1}^{s+1} - \overline{x}_{t+1}^{s+1}\|^2 + \langle x_{t+1}^{s+1} - \overline{x}_{t+1}^{s+1}, \nabla f(x_t^{s+1}) - v_t^{s+1}\rangle\Bigg]. \tag{36}$$

Using Cauchy-Schwarz and Young's inequality and the fact that $\eta \leq 1/5L$, we have the following:

$$\mathbb{E}[F(x_{t+1}^{s+1})] \tag{37}$$

$$\leq \mathbb{E}\left[F(x_t^{s+1}) + \left[\frac{L}{2} - \frac{1}{2\eta}\right]\|x_{t+1}^{s+1} - x_t^{s+1}\|^2 - \frac{\mu\eta}{3}[F(x_t^{s+1}) - F(x^*)] + \frac{\eta}{2}\|\nabla f(x_t^{s+1}) - v_t^{s+1}\|^2\right]$$

$$\leq \mathbb{E}\left[F(x_t^{s+1}) + \left[\frac{L}{2} - \frac{1}{2\eta}\right]\|x_{t+1}^{s+1} - x_t^{s+1}\|^2 - \frac{\mu\eta}{3}[F(x_t^{s+1}) - F(x^*)] + \frac{\eta L^2}{2b}\|x_t^{s+1} - \tilde{x}^s\|^2\right]. \tag{38}$$

The second inequality follows from Lemma 3. We use the similar proof technique as in Theorem 6 and 7 and define the following lyapunov function: $R_{t+1}^{s+1} = \mathbb{E}[F(x_{t+1}^{s+1}) + c_{t+1}\|x_{t+1}^{s+1} - \tilde{x}^s\|^2]$. Let $\beta = b/n$. Using the bound on the lyapunov function in Equation (15), we have the following:

$$R_{t+1}^{s+1} \leq \mathbb{E}\Bigg[F(x_t^{s+1}) - \frac{\mu\eta}{3}[F(x_t^{s+1}) - F(x^*)] + \left[c_{t+1}\left(1 + \frac{1}{\beta}\right) + \frac{L}{2} - \frac{1}{2\eta}\right]\|x_{t+1}^{s+1} - x_t^{s+1}\|^2$$

$$+ \left[c_{t+1}(1 + \beta) + \frac{\eta L^2}{2b}\right]\|x_t^{s+1} - \tilde{x}^s\|^2\Bigg]$$

$$\leq \mathbb{E}\left[F(x_t^{s+1}) - \frac{\mu\eta}{3}[F(x_t^{s+1}) - F(x^*)] + \left[c_{t+1}(1 + \beta) + \frac{\eta L^2}{2b}\right]\|x_t^{s+1} - \tilde{x}^s\|^2\right]$$

$$= R_t^{s+1} - \frac{\mu\eta}{3}\mathbb{E}[F(x_t^{s+1}) - F(x^*)]. \tag{39}$$

The second inequality follows from the fact that:

$$c_{t+1}\left(1 + \frac{1}{\beta}\right) + \frac{L}{2} \leq \frac{1}{2\eta}.$$

This, again, follows from argument stated in Theorem 6 and the fact that $\eta = b^{2/3}/(5Ln)$. Adding Equation (39) across all the iterations epoch $s+1$ and then using telescopy sum, we get

$$R_m^{s+1} \leq R_0^{s+1} - \sum_{t=0}^{m-1} \frac{\mu\eta}{3} \mathbb{E}[F(x_t^{s+1}) - F(x^*)]. \tag{40}$$

We observe that $R_m^{s+1} = \mathbb{E}[F(x_m^{s+1})] = \mathbb{E}[F(\tilde{x}^{s+1})]$. This is due the fact that $c_m = 0$ and the definition of $\tilde{x}^{s+1}$. Furthermore, $R_0^{s+1} = \mathbb{E}[F(x_0^{s+1})] = \mathbb{E}[F(\tilde{x}^s)]$. This is due to the fact that $x_0^{s+1} = \tilde{x}^s$. Therefore, using the above reasoning in Equation (40), we have

$$\mathbb{E}[F(\tilde{x}^{s+1})] \leq \mathbb{E}[F(\tilde{x}^s)] - \sum_{t=0}^{m-1} \frac{\mu\eta}{3} \mathbb{E}[F(x_t^{s+1}) - F(x^*)].$$

Adding the inequality stated above across all the epochs and using telescopy sum, we have:

$$\sum_{s=0}^{S} \sum_{t=0}^{m-1} \frac{\mu\eta}{3} \mathbb{E}[F(x_t^{s+1}) - F(x^*)] \leq \mathbb{E}[F(x^0)] - \mathbb{E}[F(\tilde{x}^S)] \leq \mathbb{E}[F(x^0)] - F(x^*).$$

The second inequality follows from the optimality of $x^*$. Using the definition of $x^k$ in PL-SVRG, we have the following:

$$\mathbb{E}[F(x^1) - F(x^*)] \leq \frac{3\mathbb{E}[F(x^0) - F(x^*)]}{\mu\eta T}$$

$$\leq \frac{\mathbb{E}[F(x^0) - F(x^*)]}{2}.$$

The second inequality follows from the fact that $T = 30Ln/\mu b^{3/2}$. Using this recursion, we have the desired result. □

*Proof.* The proof follows immediately from Theorem 8 with $b = n^{2/3}$. □

### C.3 PL-SAGA Convergence Analysis

**Theorem 9.** *Suppose $F$ is a $\mu$-PL function. Let $b \leq n^{2/3}$, $\eta = b^{3/2}/(5Ln)$ and $T = \lceil 30Ln/\mu b^{3/2} \rceil$. Then for the output $x^K$ of PL-SAGA, the following holds:*

$$\mathbb{E}[F(x^K) - F(x^*)] \leq \frac{[F(x^0) - F(x^*)]}{2^K},$$

*where $x^*$ is an optimal solution of Problem (1).*

*Proof.* We define the following :

$$\overline{x}^{t+1} = \mathrm{prox}_{\eta h}(x^t - \eta \nabla f(x^t)). \tag{41}$$

Similar to Theorem 8, we start with one iteration of PL-SAGA algorithm. In particular, we first analyze the case of $T$ iterations of SAGA. Further recursing on the the result obtain will give us the desired result. The first part of the theorem is similar to the proof in Theorem 8. Using essentially a similar argument as the one in Theorem 8 until Equation (32), we have the following:

$$\mathbb{E}[F(x^{t+1})] \leq \mathbb{E}\left[F(x^t) + \left[\frac{L}{2} - \frac{1}{2\eta}\right] \|x^{t+1} - x^t\|^2 - \frac{\mu\eta}{3}[F(x^t) - F(x^*)] + \frac{\eta L^2}{2nb} \sum_{i=1}^{n} \|x^t - \alpha_i^t\|^2\right]. \tag{42}$$

We the following Lyapunov function:

$$R_t = \mathbb{E}[F(x^t) + \frac{c_t}{n} \sum_{i=1}^{n} \|x^t - \alpha_i^t\|^2],$$

Figure 3: Non-negative principal component analysis. Performance of SGD, PROXSVRG and PROXSAGA on 'real-sim' (left), 'covtype'(center) and 'ijcnn1' (right) datasets. Recall that the y-axis is the function suboptimality i.e., $f(x) - f(\hat{x})$ where $\hat{x}$ represents the best solution obtained by running gradient descent for long time and with multiple restarts.

as defined in Theorem 7. Using the same argument in Theorem 7 to bound it, we have the following:

$$
\begin{aligned}
R_{t+1} \leq & \ \mathbb{E}\Big[F(x^t) - \tfrac{\mu\eta}{3}[F(x^t) - F(x^*)] + \Big[c_{t+1}\Big(1 + \tfrac{1-p}{\beta}\Big) + \tfrac{L}{2} - \tfrac{1}{2\eta}\Big]\|x^{t+1} - x^t\|^2 \\
& + \Big[\tfrac{c_{t+1}(1+\beta)(1-p)}{n} + \tfrac{\eta L^2}{2nb}\Big]\sum\nolimits_{i=1}^{n}\|x^t - \alpha_i^t\|^2\Big] \\
\leq & \ R_t - \tfrac{\mu\eta}{3}\mathbb{E}[F(x^t) - F(x^*)].
\end{aligned}
\tag{43}
$$

Recall that $p = 1 - (1 - 1/N)^b$. The second inequality is due to the following inequality:

$$
c_{t+1}\left(1 + \frac{1}{\beta}\right) + \frac{L}{2} \leq \frac{1}{2\eta}.
$$

This is obtained by the same argument in Theorem 7. Adding Equation (43) over all the iterations and using telescopy sum, we have the following:

$$
\mathbb{E}[F(x^T)] \leq \mathbb{E}[F(x^0)] - \sum_{t=0}^{T-1}\frac{\mu\eta}{3}\mathbb{E}[F(x^t) - F(x^*)].
$$

The above inequality is obtained from the fact that $R_T = \mathbb{E}[F(x^T)]$. This is due the fact that $c_T = 0$. Furthermore, $R_0 = \mathbb{E}[F(x^0)]$. This is due to the fact that $\alpha_i^0 = x^0$ for all $i \in [n]$. Therefore, we have:

$$
\sum_{t=0}^{T-1}\frac{\mu\eta}{3}\mathbb{E}[F(x^t) - F(x^*)] \leq \mathbb{E}[F(x^0)] - \mathbb{E}[F(x^T)] \leq \mathbb{E}[F(x^0)] - F(x^*)].
$$

Using the definition of $x^k$ in PL-SAGA, we have the following:

$$
\begin{aligned}
\mathbb{E}[F(x^1) - F(x^*)] &\leq \frac{3\mathbb{E}[F(x^0) - F(x^*)]}{\mu\eta T} \\
&\leq \frac{\mathbb{E}[F(x^0) - F(x^*)]}{2}.
\end{aligned}
$$

The second inequality follows from the fact that $T = 30Ln/\mu b^{3/2}$. Using the above recursion repeatedly, we obtain the desired result. $\qquad\square$

## D  Additional Experiments

We present the additional experiments for non-negative principal component analysis problems in this section. Figure 3 shows the additional results. Similar to Figure 2, we see that PROXSVRG and PROXSAGA outperform PROXSGD. We did not find any significant difference in the performance of PROXSVRG and PROXSAGA.

# E Lemmatta

We first few intermediate results that are useful for our analysis. These results are key to the mirror descent analysis [15]. We prove them here for completeness.

**Lemma 1.** *Suppose we define the following:*

$$y = \text{prox}_{\eta h}(x - \eta d').\tag{44}$$

*for some $d' \in \mathbb{R}^d$. Then for $y$, the following inequality holds:*

$$h(y) + \langle y - z, d' \rangle \leq h(z) + \frac{1}{2\eta}\left[\|z - x\|^2 - \|y - x\|^2 - \|y - z\|^2\right].\tag{45}$$

*for all $z \in \mathbb{R}^d$.*

*Proof.* From Lemma 5 applied on Equation 44, we get the following:

$$
\begin{aligned}
&h(y) + \langle y - x, d' \rangle + \frac{1}{2\eta}\|y - x\|^2 + \frac{\eta}{2}\|d'\|^2 \\
&= h(y) + \frac{1}{2\eta}\|y - (x - \eta d')\|^2 \\
&\leq h(z) + \frac{1}{2\eta}\|z - (x - \eta d')\|^2 - \frac{1}{2\eta}\|y - z\|^2 \\
&= h(z) + \langle z - x, d' \rangle + \frac{1}{2\eta}\|z - x\|^2 + \frac{\eta}{2}\|d'\|^2 - \frac{1}{2\eta}\|y - z\|^2.
\end{aligned}\tag{46}
$$

By rearranging Equation (46), we obtain the following inequality that concludes the proof.

$$h(y) + \langle y - z, d' \rangle \leq h(z) + \frac{1}{2\eta}\left[\|z - x\|^2 - \|y - x\|^2 - \|y - z\|^2\right].$$

$\square$

The following key lemma involving function $F$ is useful for proving the convergence of PROXSVRG and PROXSAGA.

**Lemma 2.** *Suppose we define the following:*

$$y = \text{prox}_{\eta h}(x - \eta d').$$

*for some $d' \in \mathbb{R}^d$. Then for $y$, the following inequality holds:*

$$
\begin{aligned}
F(y) \leq F(z) &+ \langle y - z, \nabla f(x) - d' \rangle \\
&+ \left[\frac{L}{2} - \frac{1}{2\eta}\right]\|y - x\|^2 + \left[\frac{L}{2} + \frac{1}{2\eta}\right]\|z - x\|^2 - \frac{1}{2\eta}\|y - z\|^2.
\end{aligned}\tag{47}
$$

*for all $z \in \mathbb{R}^d$.*

*Proof.* We have the following inequalities for function $f$:

$$f(y) \leq f(x) + \langle \nabla f(x), y - x \rangle + \frac{L}{2}\|y - x\|^2,$$

$$f(x) \leq f(z) + \langle \nabla f(x), x - z \rangle + \frac{L}{2}\|x - z\|^2.$$

The above inequalities can be obtained by application of Lemma 6. Adding both the inequalities above, we obtain the following inequality:

$$f(y) \leq f(z) + \langle \nabla f(x), y - z \rangle + \frac{L}{2}\left[\|y - x\|^2 + \|z - x\|^2\right].\tag{48}$$

Adding Equations (45) (which follows from Lemma 1) and (48), we obtain the inequality:

$$
\begin{aligned}
F(y) \leq F(z) &+ \langle y - z, \nabla f(x) - d' \rangle \\
&+ \left[\frac{L}{2} - \frac{1}{2\eta}\right]\|y - x\|^2 + \left[\frac{L}{2} + \frac{1}{2\eta}\right]\|z - x\|^2 - \frac{1}{2\eta}\|y - z\|^2.
\end{aligned}\tag{49}
$$

Here we used that definition $F(x) = f(x) + h(x)$. This concludes our proof. $\square$

The following result is useful for bounding the variance of the updates of PROXSVRG and follows from slight modification of result in [21]. We give the proof here for completeness.

**Lemma 3** ([21]). *For the iterates $x_t^{s+1}, v_t^{s+1}$ and $\tilde{x}^s$ where $t \in \{0, \ldots, m-1\}$ and $s \in \{0, \ldots, S-1\}$ in Algorithm 1, the following inequality holds:*

$$\mathbb{E}[\|\nabla f(x_t^{s+1}) - v_t^{s+1}\|^2] \leq \frac{L^2}{b} \mathbb{E}\|x_t^{s+1} - \tilde{x}^s\|^2.$$

*Proof.* Let $I_t = \{i_1, \cdots, i_b\}$. Let us define the following notation for the ease of exposition:

$$\zeta_t^{s+1} = \frac{1}{b} \sum_{k=1}^{b} \left( \nabla f_{i_k}(x_t^{s+1}) - \nabla f_{i_k}(\tilde{x}^s) \right).$$

Also, let $\zeta_{t,i}^{s+1} = \nabla f_i(x_t^{s+1}) - \nabla f_i(\tilde{x}^s)$. Using this notation, we obtain the following bound:

$$\mathbb{E}_{I_t}[\|\nabla f(x_t^{s+1}) - v_t^{s+1}\|^2] = \mathbb{E}_{I_t}[\|\zeta_t^{s+1} + \nabla f(\tilde{x}^s) - \nabla f(x_t^{s+1})\|^2]$$

$$= \mathbb{E}_{I_t}[\|\zeta_t^{s+1} - \mathbb{E}_{I_t}[\zeta_t^{s+1}]\|^2] = \frac{1}{b^2} \mathbb{E}_{I_t} \left[ \left\| \sum_{k=1}^{b} \left( \zeta_{t,i_k}^{s+1} - \mathbb{E}_{i_k}[\zeta_{t,i_k}^{s+1}] \right) \right\|^2 \right]$$

The second equality is due to the fact that $\mathbb{E}_{I_t}[\zeta_t^{s+1}] = \mathbb{E}_{i_k}[\zeta_{t,i_k}^{s+1}] = \nabla f(x_t^{s+1}) - \nabla f(\tilde{x}^s)$. From the above relationship, we get

$$\mathbb{E}_{I_t}[\|\nabla f(x_t^{s+1}) - v_t^{s+1}\|^2] = \frac{1}{b^2} \left[ \sum_{k=1}^{b} \mathbb{E}_{i_k} \|\zeta_{t,i_k}^{s+1} - \mathbb{E}_{i_k}[\zeta_{t,i_k}^{s+1}]\|^2 \right]$$

$$\leq \frac{1}{b^2} \left[ \sum_{k=1}^{b} \mathbb{E}_{i_k} \|\nabla f_{i_k}(x_t^{s+1}) - \nabla f_{i_k}(\tilde{x}^s)\|^2 \right] \leq \frac{L^2}{b} \|x_t^{s+1} - \tilde{x}^s\|^2.$$

The first equality follows from Lemma 7. The first inequality is due to the fact that for a random variable $\zeta$, $\mathbb{E}[\|\zeta - \mathbb{E}[\zeta]\|^2] \leq \mathbb{E}[\|\zeta\|^2]$. The last inequality follows from $L$-smoothness of $f_i$. $\square$

A similar result can be obtained for PROXSAGA. The key difference with that of Lemma 3 is that the variance term in PROXSAGA involves $\alpha_i^t$. Again, we provide the proof for completeness.

**Lemma 4.** *For the iterates $x^t, v^t$ and $\{\alpha_i^t\}_{i=1}^n$ where $t \in \{0, \ldots, T-1\}$ in Algorithm 2, the following inequality holds:*

$$\mathbb{E}[\|\nabla f(x^t) - v^t\|^2] \leq \frac{L^2}{nb} \sum_{i=1}^{n} \mathbb{E}\|x^t - \alpha_i^t\|^2.$$

*Proof.* Let $I_t = \{i_1, \cdots, i_b\}$. As before, we define the following notation for the ease of exposition:

$$\zeta_t = \frac{1}{b} \sum_{k=1}^{b} \left( \nabla f_{i_k}(x^t) - \nabla f_{i_k}(\alpha_{i_k}^t) \right).$$

Also, let $\zeta_{t,i_k} = \nabla f_{i_k}(x^t) - \nabla f_{i_k}(\alpha_{i_k}^t)$. In this notation, we have the following:

$$\mathbb{E}_{I_t}[\|\nabla f(x^t) - v^t\|^2] = \mathbb{E}_{I_t} \left[ \left\| \zeta_t + \frac{1}{n} \sum_{i=1}^{n} \nabla f(\alpha_i^t) - \nabla f(x^t) \right\|^2 \right]$$

$$= \mathbb{E}_{I_t}[\|\zeta_t - \mathbb{E}[\zeta_t]\|^2] = \frac{1}{b^2} \mathbb{E}_{I_t} \left[ \left\| \sum_{k=1}^{b} \left( \zeta_{t,i_k} - \mathbb{E}_{i_k}[\zeta_{t,i_k}] \right) \right\|^2 \right]$$

The second equality follows from the fact that $\mathbb{E}_{I_t}[\zeta_t] = \mathbb{E}_{i_k}[\zeta_{t,i_k}] = \nabla f(x^t) - \frac{1}{n}\sum_{i=1}^{n}\nabla f(\alpha_i^t)$. From the above inequality, we get

$$\mathbb{E}_{I_t}[\|\nabla f(x^t) - v^t\|^2] = \frac{1}{b^2}\left[\sum_{k=1}^{b}\mathbb{E}_{i_k}\|\zeta_{t,i_k} - \mathbb{E}_{i_k}[\zeta_{t,i_k}]\|^2\right]$$

$$\leq \frac{1}{b^2}\left[\sum_{k=1}^{b}\mathbb{E}_{i_k}\|\nabla f_{i_k}(x^t) - \nabla f_{i_k}(\alpha_{i_k}^t)\|^2\right] \leq \frac{L^2}{nb}\sum_{i=1}^{n}\|x^t - \alpha_i^t\|^2.$$

The first equality is due to Lemma 7. The first inequality follows from the fact that for a random variable $\zeta$, $\mathbb{E}[\|\zeta - \mathbb{E}[\zeta]\|^2] \leq \mathbb{E}[\|\zeta\|^2]$. The last inequality is from $L$-smoothness of $f_i$ for all $i \in [n]$ and uniform randomness of the set $I_t$. □

The following lemma is a classical result in mirror descent analysis.

**Lemma 5.** *Suppose function $h : \mathbb{R}^d \to \mathbb{R}$ is l.s.c and $y = \mathrm{prox}_{\eta h}(x)$. Then we have the following:*

$$h(y) + \tfrac{1}{2\eta}\|y - x\|^2 \leq h(z) + \tfrac{1}{2\eta}\|z - x\|^2 - \tfrac{1}{2\eta}\|y - z\|^2,$$

*for all $z \in \mathbb{R}^d$.*

**Lemma 6.** *Suppose function $f : \mathbb{R}^d \to \mathbb{R}$ is $L$-smooth, then we have the following:*

$$f(y) + \langle \nabla f(y), x - y \rangle - \frac{L}{2}\|x - y\|^2 \leq f(x) \leq f(y) + \langle \nabla f(y), x - y \rangle + \frac{L}{2}\|x - y\|^2,$$

*for all $x, y \in \mathbb{R}^d$.*

**Lemma 7.** *For random variables $z_1, \ldots, z_r$ are independent and mean 0, we have*

$$\mathbb{E}\left[\|z_1 + \ldots + z_r\|^2\right] = \mathbb{E}\left[\|z_1\|^2 + \ldots + \|z_r\|^2\right].$$

*Proof.* We have the following:

$$\mathbb{E}\left[\|z_1 + \ldots + z_r\|^2\right] = \sum_{i,j=1}^{r}\mathbb{E}\left[z_i z_j\right] = \mathbb{E}\left[\|z_1\|^2 + \ldots + \|z_r\|^2\right].$$

The second equality follows from the fact that $z_i$'s are independent and mean 0. □

**Lemma 8.** *For random variables $z_1, \ldots, z_r$, we have*

$$\mathbb{E}\left[\|z_1 + \ldots + z_r\|^2\right] \leq r\mathbb{E}\left[\|z_1\|^2 + \ldots + \|z_r\|^2\right].$$