[Reviews · NeurIPS 2016]

Reviewer 1

Summary

A good paper on randomized incremental gradient methods for non-convex problems.

Qualitative Assessment

The authors did an excellent job in extending randomized incremental gradient methods to non-convex composite optimization. Their key contribution was to show that the size of mini-batch should be dependent on the number of terms in order to gain an O(1/n^{1/3}) factor in the complexity bound. The authors should point out to the readers that the SGD method for non-convex optimization was designed for solving stochastic optimization problems possibly with continuous random variables and/or for solving the generalization (rather than empirical) risk minimization in machine learning. Therefore, the numerical results seem to be a bit biased towards SAGA or SVRG since the error measure was empirical. To have a fair comparison, the authors should at least make this point clear in both the introduction and numerical experiment parts

Confidence in this Review

2-Confident (read it all; understood it all reasonably well)


Reviewer 2

Summary

The paper studies stochastic algortihms for minimizing a regularized finite sum problem, where the regularizer is nonsmoth but convex; and the functions in the finite sum are smooth but possibly nonconvex. This problem is not well studied and hence understood from a complexity point of view. Previous known results give complexity bounds (to a stationary point defined using the notion of the proximal mapping) for a method with increasing minibatch sizes. The present paper sets out to give the first results which do not require growing minibatches. Linear convergence results are obtained under a Polyak-Łojasiewicz condition. Minibatch proximal versions of the SAGA and SVRG methods are proposed and studied.

Qualitative Assessment

The paper is a pleasure to read – excellent writing. The ideas are well explained, the paper proceeds smoothly. The paper is well set in the context of existing results and literature (one small issue is the omission to mention that a minibatch version of Prox SVRG was analysed before in the case of smooth losses – and is known as the mS2GD method; see line 124). The complexity (in terms of number of stochastic gradient evaluations; IFO) for both is shown to be O(n + n^{2/3}/eps); which is better than O(1/eps^2) rate of ProxSGD and O(n/eps) rate of ProxGD. The complexity is also improved for PL functions. The new results are a good addition to our knowledge of complexity of stochastic methods for an important class of nonconvex problems relevant to ML. Some previous results are included as special cases of this approach (e.g., [27], [34]). Experiments with non-negative PCA show these methods work about the same as much better than ProxSGD. Q: Are any lower bounds known for the IFO complexity for this problem? Remark: The word “fast” in the title evokes the use of Nesterov’s “acceleration” – but this is not the case. It is tempting to call a method fast, but if this is to be justified, either extensive computational experiments should be performed against all or most competing methods; or the word should have a special meaning (such as Nesterov’s acceleration). I suggest the word be dropped from the title and replaced by “Complexity of”. Small issues: a) Line 40: to -> to be b) Table 1 contains complexity results for proxSGD under PL assumption. Where have these been proved (citation is missing)? Confusingly, the caption says that no specific complexity results exist. Do they exist or not? c) Line 151: For the -> For d) Line 154: is a -> be a e) Line 236: worser -> worse f) References: [9] : capitalize P and L in PL; [22] appeared in ICML 2015; [25] – capitalize journal name; [26] appeared in Math Prog 2016; [33] appeared; [34] – write SVRG and not svrg. ----- post-rebuttal comments ----- I have read the rebuttal. I have the same opinion of the paper as before and keep my scores.

Confidence in this Review

3-Expert (read the paper in detail, know the area, quite certain of my opinion)


Reviewer 3

Summary

This paper analyzes two types of stochastic procedures for solving a very important and general class of optimization problems. In particular, the objective function is a superposition of a non-smooth convex function and another non-convex smooth function. The paper focuses on two important quantities: (1) PO complexity, and (2) IFO complexity. The authors show that the algorithms can provably converge to some stationary points with the desired PO and IFO complexities, in the presence of constant mini-batches. The paper is very well-written and is a pleasure to read. I believe that developing fast stochastic procedures is an important contribution for solving many nonconvex problems. The only part that I think can be improved is to provide more interpretations regarding the particular parametric choices used in the theorems (like m and b). It would be nice to discuss whether a broader range of parameter choices can also lead to similar performance, either theoretically or practically.

Qualitative Assessment

1. Theorem 2 chooses b = n^{2/3} and m = n^{1/3} and obtains intriguing performance guarantees. But these two choices of parameters seem a bit mysterious to me. Can the authors provide some intuition / interpretation for such choices of b and m? Can we find a wider range of b and m that can also lead to the desired IFO and PO complexities? 2. It would be good to define or explain the parameters T and b in Algorithm 1. 3. Explain the parameter "epoch length" for the benefits of those readers who are not familiar with this subject. In general, how does it affect the performance? 4. For ProxSAGA, can the authors provide some brief interpretation regarding the analytical difficulty of using just one set (as opposed to two independent sets I_t and J_t)? Which version (one set vs. two sets) performs better in practice? 5. Similar to Comment 1, why does Theorem 4 set b=n^{2/3}? Any particular reason? And is the result sensitive to this particular choice?

Confidence in this Review

1-Less confident (might not have understood significant parts)


Reviewer 4

Summary

The author analyse the rate of convergence of stochastic algorithm with constant minibatch size to a stationary point, in the case where the function is non-smooth and non-convex. They also analyse the convergence rate when the function satisfies the PL-inequality (can be compared to strongly convex functions in the convex case).

Qualitative Assessment

Overall, the article is well written and clear. They explain clearly what are the contributions of their work. However, even if the theoretical analysis is complete, the impact of the results as well as the numerical experiments are not convincing. The authors shows the convergence of stochastic algorithm using constant-size minimatches, but the (optimal) size must be of the order of n^{2/3}, which is quite big and comparable to the size of the deterministic gradient method. Moreover, the comparison (deterministic vs stochastic) is not done in the numerical experiments section. Also, the algorithms are unchanged, they just adapted the size of the minibatch to have theoretical guarantee (but we can not deny that this case is more realistic than having a size-varying minibatch). Also, in the experiments section the authors use a batch size of 1 but they proved that the size n^{2/3} must be better. Moreover, the starting point is too close to the optimal point (f(x0)-f(x*) ~= 10^{-3}, which is an acceptable accuracy in machine learning). They should start from less accurate starting points (Especially since the authors insists they proved non-asymptotic rate of convergence, unlike in previous works). A few other remarks: - Parameter $m$ not defined - The used quantities, e.g. $m$, $T$, etc..., should be explained (in an intuitive way if possible). - The title is way too general ("nonsmooth non-convex optimization") and should be changed (for example, by including the fact that it uses proximal operator or by mentioning the sum of smooth functions). - The captions in the figures are too small. In conclusion: - The article is very clear and the contributions are well presented. - The theoretical analysis is complete and the theorems are easily understandable. - The ideas are not original, since people already use constant minibatch for practical problem. However it gives guarantees about the performances of this method. - The theoretical analysis shows the convergence for constant size minibatch, but the size is much too large for practical algorithms. - The numerical analysis is not consistent with the theoretical results (it uses minibatch size of 1 instead of n^{2/3}, starting point too close to the optimal point). - Since the batch size is big, there is no comparison with deterministic gradient method.

Confidence in this Review

1-Less confident (might not have understood significant parts)


Reviewer 5

Summary

This paper analyzed the convergence of proximal-SVRG and proximal-SAGA for the non-convex problems. Then their analysis was extent to Polyak-Lojasiewicz functions. Last, some numerical experiments were conducted on non-negative PCA problem.

Qualitative Assessment

This paper is well written. Theoretical analysis and numerical experiments are solid. However, I personally think that providing a convergence bound for the proximal versions of some existing methods is enough, considering NIPS is so competitive now. Though I did not check the proof line by line, I feel the analyzing method mainly follows [22] and [23] and so there is not too much new. This is why I give a low score concerning novelty.

Confidence in this Review

3-Expert (read the paper in detail, know the area, quite certain of my opinion)


Reviewer 6

Summary

This paper tackles the challenging problem of minimizing nonconvex, nonsmooth, finite-sum objective functions. The authors adapted the analysis of two recent SGD-based algorithms with variance reduction, namely SVRG and SAGA, to the search of stationary points in the nonconvex case. They provide theoretical choice for the mini-batch size to ensure faster convergence. They also considered a particular case where the convergence is linear, and gave experimental results.

Qualitative Assessment

The article is overall interessant. The introduction is clear and sums all the recent results proved in the domain. It also gives insight about the differences from the convex setting to the nonconvex setting. My main comment is for Corollary 2 and Corollary 4 about the IFO complexity. Its computation is done too fast and it hides an error. Let's focus on lines 158 to 167. I'm using the notations of Algorithm 1. - First comment: the IFO complexity of one inner loop equals b*m + n. But you have said on Theorem 2 that b = n^{2/3} and m = n^{1/3}, so that the product b*m = n. Then the total IFO complexity should write O( n^{2/3} / \epsilon ). - Second comment: I'm ok with the part O( n^{2/3} / \epsilon ) in the IFO complexity. However, for the second part, the complexity writes S*n=(T/m)*n, and you said on line 165 that since "T is a multiple of m", you can write O( (T/m)*n ) = O( n ), which is not true. T/m may still be large, image the case where T = m * n. Keeping the dependency on n, and \epsilon, the second term writes O( 1 / \epsilon * n^{-1/3} * n ) = O( n^{2/3} / \epsilon ), which equals the former. These comments doesn't alter the quality of the bounds proved by the authors. But this wrong complexity occurs many times in the article. Also, I've noticed a self-reference on line 95: "We build upon...".

Confidence in this Review

2-Confident (read it all; understood it all reasonably well)